# Analysis of Arbuscular Mycorrhizal Fungal Inoculant Benchmarks

**DOI:** 10.3390/microorganisms9010081

**Published:** 2020-12-31

**Authors:** Sulaimon Basiru, Hopkins Pachalo Mwanza, Mohamed Hijri

**Affiliations:** 1African Genome Center—AgroBioSciences, Mohammed VI Polytechnic University (UM6P), Lot 660, Hay Moulay Rachid, Ben Guerir 43150, Morocco; Sulaimon.BASIRU@um6p.ma (S.B.); Hopkins.MWANZA@um6p.ma (H.P.M.); 2Institut de Recherche en Biologie Végétale, Département de sciences biologiques, Université de Montréal, 4101 Sherbrooke Est, Montréal, QC H1X 2B2, Canada

**Keywords:** arbuscular mycorrhizal fungi, bioinoculants, biostimulants, biofertilizers, crop production, inoculant formulation, field applications

## Abstract

Growing evidence showed that efficient acquisition and use of nutrients by crops is controlled by root-associated microbiomes. Efficient management of this system is essential to improving crop yield, while reducing the environmental footprint of crop production. Both endophytic and rhizospheric microorganisms can directly promote crop growth, increasing crop yield per unit of soil nutrients. A variety of plant symbionts, most notably the arbuscular mycorrhizal fungi (AMF), nitrogen-fixing bacteria, and phosphate-potassium-solubilizing microorganisms entered the era of large-scale applications in agriculture, horticulture, and forestry. The purpose of this study is to compile data to give a complete and comprehensive assessment and an update of mycorrhizal-based inoculant uses in agriculture in the past, present, and future. Based on available data, 68 mycorrhizal products from 28 manufacturers across Europe, America, and Asia were examined on varying properties such as physical forms, arbuscular mycorrhizal fungal composition, number of active ingredients, claims of purpose served, mode of application, and recommendation. Results show that 90% of the products studied are in solid formula—powder (65%) and granular (25%), while only 10% occur in liquid formula. We found that 100% of the products are based on the Glomeraceae of which three species dominate among all the products in the order of *Rhizophagus irregularis* (39%), *Funneliformis mosseae* (21%), *Claroideoglomus etunicatum* (16%). *Rhizophagus clarus* is the least common among all the benchmark products. One third of the products is single species AMF and only 19% include other beneficial microbes. Of the sampled products, 44% contain AMF only while the rest are combined with varying active ingredients. Most of the products (84%) claimed to provide plant nutrient benefits. Soil application dominates agricultural practices of the products and represents 47%. A substantial amount of the inoculants were applied in cereal production. Recommended application doses varied extensively per plant, seed and hectare. AMF inoculant seed coating accounted for 26% of the products’ application and has great potential for increased inoculation efficiency over large-scale production due to minimum inoculum use. More applied research should also be conducted on the possible combination of AMF with other beneficial microbes.

## 1. Introduction

There has been a remarkable surge in development of the mycorrhizal-based inoculants market in the last two decades, essentially in horticulture and field crop production. The biofertilizer market in agriculture is estimated to reach USD 2.3 billion by 2022, at a Compound Annual Growth Rate (CAGR) of 14.08% during this period [1]. Arbuscular mycorrhizal fungi (AMF) are especially used in most bioinoculant production as they have been known for establishing symbiotic relationships with more than 85% of plant species of agricultural interest [2]. They have been linked to several benefits including macro- and micro-nutrient uptake, water absorption, soil aggregate stability [3,4], salinity and drought stress suppression, trace metal detoxification, and protection against pathogens and herbivores [5]. AMF provides these numerous benefits to plants in exchange for carbohydrates and other photosynthetic derivatives [6].

Several reasons are attributed to the unprecedented booming of the mycorrhizal inoculant industry. There is a growing scientific evidence proving various benefits provided to crops by the mycorrhizal inoculants in terms of growth and yield, which has attracted much interest from end users. Rising global population with corresponding food demand and a growing concern for the environment has also increased the need for bio-fertilization. The United Nations predicted that the global population would increase from the current 7.7 billion to 8.5 billion in 2030 and to 9.7 billion in 2050 [7] with food demand forecasted to rise by 70% in 2050 [8]. Agricultural intensification is the main solution to overcome impending food crisis, but also constitutes corresponding threat to the environment. Therefore, there is need for paradigm shift to a sustainable agricultural production system that advocates for environmentally friendly practices [9]. Biofertilizers, especially arbuscular mycorrhizal fungi, are becoming an integral part of sustainable agriculture. They have been recognized as ecologically and economically important, performing the roles of fertilizers and pesticides [5]. AMF have become a key component of organic farming and have contributed to the success of the farming system by maintaining long-term soil health and fertility [10,11]. AMF are increasingly dominating the biofertilizer market space and have proven to be practicable options to improve crop productivity [12].

According to Berruti et al. (2015) [6], AMF inoculation can provide immense benefits by cutting down expenses for growers, and land recovery projects. It has been observed that mycorrhizal-based products are more cost-effective than conventional fertilizers especially in regions where phosphorus depletion in soils is a serious plant nutrition problem, thus driving the demand for large scale production. For instance, in India, commercial mycorrhizal-based inoculants are being widely used in rice production to thwart the effects of low phosphorus levels in the soil combined with the rising cost of synthetic P fertilizers.

Despite these benefits and justifiable reasons for the market explosion, only a few AMF strains are marketed globally as biofertilizers because of significant limitations hindering mass production of AMF inoculants given their obligate biotrophic lifecycle, meaning that they require a host plant to grow and reproduce. In addition, there is great concern about the quality and quantity of the mycorrhizal inoculants [13,14]. Efficacy of inoculants is also affected by the different field conditions such as compatibility to various soil characteristics, different crop species, indigenous microbial communities, and environmental factors as well as the soil fertility management practices of the native soils [15,16]. These constraints influence soil microbial dynamics and functional processes impacting the performance of commercial bioinoculants. Consequently, there has been a conflicting stance on their efficacy in field conditions [17]. Owen et al. (2015) [18] mentioned that there was a distinct lack of robust field-based testing of commercial bioinoculants as most studies have focused on greenhouse pot trials. Therefore, for AMF inoculant industry to thrive, rigorous research must be conducted to provide best practices to the inoculant companies regarding composition, quality, quantity, and application methods of the products [19]. Although lack of efficacy or negative impacts of AMF inoculation has been reported [20,21,22], recent studies conducted under field conditions have shown promising results [12,23,24,25,26].

A growing number of market players are investing in mycorrhizal inoculant production, but very limited information is publicly available regarding commercial inoculants in the market. In this study, we collected and analyzed important data available on benchmark mycorrhizal products to synthesize and compare their characteristics such as composition, formulation of products, propagule contents, claimed crop benefits, active ingredients, and mode and types of applications. We also reviewed recent trials of commercial AMF inoculants under greenhouse and field conditions. Lastly, we revealed the *status quo* in the AMF inoculant markets and industry and share our perspectives on potential market-based research opportunities.

## 2. Materials and Methods

### 2.1. Data Collection

Data were collected on 68 products from 28 different manufactures located in nine countries: USA, Canada, Spain, Israel, The Netherlands, Mexico, Chile, Italy, and France. We gathered technical information from official publicly available data on the companies’ website or by contacting the company via electronic or physical means where the information was not publicly available. The main sources of data included official websites, labels/MSDSs, and regulatory databases. We assessed the data for physical formulations, product composition, claims about potential crop benefits of the inoculants, application mode and dose recommendation. Raw data are shown in Appendix A.

### 2.2. Data Analysis

The data were gathered and analyzed using Microsoft Excel 365 (Microsoft, Redmond, WA, USA). The same data tool was used to produce graphs, charts and tables.

## 3. Results

### 3.1. Mycorrhiza Inoculant Company by Country

Data were collected from nine countries: Canada, Chile, France, Israel, Italy, Mexico, The Netherlands, Spain and USA. Five companies dominated in terms of quantity for these products: Premier Tech (Rivière-du-Loup, QC, Canada); GroundWork BioAG (Mazor, Israel); Plant Health Care (Raleigh, NC, USA); Valent; Helena Agri-Enterprises, LLC (Collierville, TN, USA); Tainio Biologicals, Inc. (Spokane, WA, USA) and Atens (Irvine, CA, USA). Of the 68 products, 20 were manufactured in the US, but the company with the largest number of products in the study was Premier Tech (Table 1).

### 3.2. Formula and Composition of Mycorrhizal Inoculants

Results indicated that inoculants exist mainly in three formulations: liquid, powder, and granular products. Over 60% of the inoculants surveyed exist in powder formula while both granular and liquid products account for approximately 25% and 10%, respectively (Appendix A).

Most of the products contain Glomeracea genera: *Rhizophagus iranicus* (syn. *Glomus Iranicum*), *Rhizophagus aggregatus* (syn. *Glomus aggregatum*), *Rhizophagus irregularis* (syn. *Glomus irregulare* previously misidentified as *Glomus intraradices*), *Funneliformis mosseae* (syn. *Glomus mosseae*), *Claroideoglomus etunicatum* (syn. *Glomus etunicatum*), *Septoglomus deserticola* (syn. *Glomus deserticola*), *Rhizophagus clarus* (syn. *Glomus clarum*). Two major species, *Rhizophagus irregularis* and *Funneliformis mosseae*, are the most common AMF species among the products; however, one third of the products contain *R. irregularis* making it the most common AMF strain, as it was found in 39% of the products followed by *F. mosseae* with 21% (Appendix A). Similarly, results showed that most of the commercial inoculants consist of AMF consortia (66%), while only a single strain inoculant is present in 34%. Nevertheless, among those that are consortia, only 22% contain between two and three AMF species, about 26% consist of four species, while only 9% have five species or more (Appendix A).

In addition to AMF, commercial inoculants often contain other active ingredients such as bacteria (for example: *Bradyrhizobium japonicum*, *Bacillus* sp., *Azospirillum brasilense*, etc.) and non-mycorrhizal fungal species (e.g., *Trichoderma* spp.), as well as bio-stimulants such as amino acids, algae extract, and humic acid. Some inoculants also contain macro- and micro-nutrients. About 44% contain only AMF, 7% contain humic acid and mineral nutrients while 19% contain other microbes including some other AMF taxa such as *Gigaspora fasciculata*, *G. geosporum*, *G. constrictum*, and *G. tortiosum*, *Entrophospora Columbiana*, as well as N-fixing and P-solubilizing bacteria (Appendix A).

### 3.3. Propagule Concentration

This study shows that most products contain more than 100 propagules per gram, with 1 to 8 different AMF species. Propagule concentration is independent of the number of AMF species. Highly concentrated products have more than 10,000 propagules per gram, in liquid or powder form, but more products in powder forms are highly concentrated. Although some products with multiple AMF species have higher propagule concentration, this is not a rule as many inoculants with higher propagule numbers were found to have fewer propagules than single-species inoculants (Appendix A).

### 3.4. Industrial Claims on Effects of the AMF-Based Inoculants

This study found that companies producing mycorrhizal-based inoculants made numerous claims associated with the potency of the inoculants on crop productivity. More than eight different claims were gathered from different products. On average, each product is claimed to perform three different roles, with nutrient benefits dominating at 84% (Figure 1). Nutrient benefits include solubilizing and facilitating uptake of minerals. Plant growth and vigor enhancement is the next claimed role of the commercial inoculants and accounted for nearly 70% of the products. Furthermore, 63% of the products help to increase resilience of crops to climate stress, about 60% of the products has the potential to alleviate soil stress, approximately 60% can improve crop quality and quantity while only 25% of the products were claimed to enhance soil microbial activity. Other benefits claimed by the manufacturers as observed in the study include water uptake, soil aggregation, establishment of seedlings, reduction of soil erosion, and protection against pathogen attacks.

### 3.5. Recommended Inoculant Application Methods and Doses

Three main categories of product application practices were recorded during the study. Most of the products were recommended for soil applications (47%) and could be applied as liquid or powder formulation.

Seed treatment was the next application practice accounting for 26% of all the products. 11% of the inoculants can be co-applied with chemical fertilizers and only 4% of the products may follow any of the three means of application (Figure 2). In this survey, application rates for powdered products ranged from 0.2 g to 10 g per plant, 0.1 g to 2 kg per kg of seeds and 0.12 kg to 30 kg per hectare, while liquid inoculants ranged from 0.2 mL to 10 mL; however, no direct correspondence was found between application rates and propagule number.

This survey further revealed that most products were applied in cereal production, with 54% of them used as starters and only 46% coated on seeds. Starters are the inoculants which are applied simultaneously with seeds during sowing. In horticulture, 42% of the inoculants are applied as starters, 16% coated on seeds (16%) and 42% were applied during transplanting of vegetable seedlings. A significant proportion of inoculant applications were observed in nurseries with 50% application as starters, 25% seed treatment and 29% during transplanting. The least application was observed in tree cultivation with 42% as starters, 33% during seedling transplanting, and 25% during flowering (Figure 3).

## 4. Discussion

### 4.1. Product Breakdown by Country of Production

Success in mycorrhiza inoculant production has reached a new phase and the industry has been growing rapidly. This survey is the first to assess the variability in commercial mycorrhizal inoculants. Our study focused on 68 products from 28 manufacturers mainly in Europe, North America, South America and Asia. Our sampling is not exhaustive and was based on information available during the study. However, there is no valid data on the actual number of firms producing inoculants, the number of propagules produced, or the number of areas treated with mycorrhizal inoculants. Our study is in line with previously published work [27,28]. For example, in 2016, Pal et al. (2016) [27] listed more than sixty manufacturers of mycorrhizal inoculants across Asia, Europe, North America, South America and South Africa. In 2018, Chen et al. [28] also identified more than seventy-five firms producing and marketing mycorrhizal-based inoculants in Europe alone, an increment over the small number of (ten to forty) firms involved in the business in 2000 and 2010, respectively (Appendix A). Major players in the mycorrhizal industry were found in the United States, Canada, Germany, Italy, Czech Republic, United Kingdom, and Spain. Reasons for this could be attributed to available data as well as to increased awareness about the environmental benefits of mycorrhizal fungi, increased demand for organic food products and the availability of modern technology for inoculant production.

Reports indicated Asia-Pacific as the third largest player (after North America and Europe) in the global biofertilizer market with increasing demand in India, China and Taiwan [1]; however, local data on AMF inoculants was unavailable for inclusion in this study. Despite a long history of AMF research in Australia, AMF is rarely considered by farmers in management decisions due to a lack of agronomic-relevant recommendations, which resulted from a lack of dialogue between AMF researchers and agronomists [29]. Thus, the status of commercial mycorrhiza production in Australia is still unclear although much literature has been published on the role of mycorrhizal fungi in crop production and forestry [29,30,31,32,33].

Mycorrhiza production in Africa is still at a medium or small scale due to technological limitations but Pal et al. (2016) [27] indicated South Africa and Kenya as being among the major players in Africa.

### 4.2. AMF-Based Inoculant Formulation

Our survey found that more than 60% of the products are in powder formulation while only 29% are in liquid. This could be explained by the methods of mass production *in vivo* versus *in vitro* as well as the conservation and stability of mycorrhizal propagules. Most of the solid inoculants contain more than one species of AMF most likely produced *in vivo* using conventional co-culture with a host plant in a substrate usually under greenhouse; however, all of the liquid inoculants are only available in one single strain *Rhizophagus irregularis* isolate DAOM197198 as the only active ingredient (Appendix A). So far, *R. irregularis* is the most successful strain that is produced *in vitro* using a large-scale mycoreactor with transformed roots [34], although many other strains have also been cultured successfully *in vitro* but at a laboratory scale [35].

Large-scale AMF inoculants are produced *in vivo* by co-cultivating AMF and host plants in inert substrates to allow for propagation, and substrates rich in propagules are harvested at the end of cycle [36]. The conventional culture technique is cheap and often leads to solid products, but the main drawback is high-risk contamination and a low concentration of AMF propagules. *In vitro* production techniques offer contamination-free propagules due to the sterile culture conditions. Products can be handled to contain more propagules with recent data indicating that propagule numbers have been increasing significantly from hundreds of spores produced initially to up to several thousand per mL [36].

Solid substrate inoculants have gained wider application possibly due to the ease of handling (mixing), conservation and richness of AMF species and inclusion of other beneficial microbes such as plant growth promoting bacteria, but application in large field is faced with some drawbacks. First major challenge with a solid substrate inoculant is the labor-intensive application especially in large-scale operation. It requires special machinery that suits different varieties of plant, soil and fertilization program to have a homogeneous inoculation process [37]. Miguel et al. (2007) [38] also reported a number of limitations with solid inoculants: spore germination is affected by long dormancy periods due to the packaging conditions; propagule concentration in a solid inoculant is also natural and cannot be increased to the desired size; and glomalin (glomalin-related soil proteins) that accelerates formation of stable soil aggregates is not excreted on solid substratum.

Introducing a liquid substrate helps to overcome some of these limitations. It reduces the dormancy period of spores; it enables the propagule concentration to be increased to the desired amount; and improves the formation of soil aggregates by stimulating glomalin secretion at high concentrations. Glomalin secretion at high amounts in liquid media earns additional marketing claims for manufacturers. Therefore, it is not surprising that most inoculants that claimed to contribute to the improving soil structure are in liquid forms (Appendix A). Liquid inoculants also allow easy handling and low transportation costs as they can be designed to contain more propagules than solid products. The liquid inoculant is better suited for fertigation and irrigation than solid inoculants [37,38].

It may be essential to highlight that the liquid inoculant is not a complete substitute of the solid-based inoculant as it does not solve all the problems. It has a short shelf life due to its liquid form and this may limit its commercial application. It is a sterile product lacking beneficial microbes compared to inoculants in solid form. The negative impact of the sterile condition and artificial way of production as well as the lack of beneficial microbes on inoculants have been reported, although there is little published data available. Calvet et al. (2013) [39] reported that *in vitro*-based inoculants produced less spores and recorded lower mycorrhization than *in vivo* inoculants in leek plants. Moreover, only *R. irregularis* is available in liquid form as confirmed by this study. It may be perceived that future demand will favor liquid inoculants due to its various advantages under field condition, but solid inoculants are well-established products, especially in forestry, gardening, and horticulture.

### 4.3. Inoculant Composition

Arbuscular mycorrhizal fungi belong to the phylum Glomeromycota which consists of four orders, 12 families, 41 genera and approximately 338 species [40]. Gigasporaceae, Glomeraceae, and Acaulosporaceae represent the most diverse genera within the phylum, containing 82% of the entire species [41]. Glomeraceae family includes the abundant genera such as *Glomus, Rhizophagus, Funneliformis* and *Septoglomus,* which have been reported in all continents [41,42,43].

Most of the commercial inoculants evaluated in the present study were found to include species belonging to *Glomus, Funneliformis, Rhizophagus and Septoglomus* genera. Among all the AMF species present in the products, *R. irregularis* and *F. mosseae* were the dominant species. *R. irregularis* was the most widespread, occurring in more than one-third of the products (Appendix A).

Members of the Glomeraceae have displayed differential efficacy in terms of host root colonization and performance under varying field conditions. For example, species of *Glomus* and *Rhizophagus* have been reported to outperform other genera such as *Gigaspora* and *Scutellospora* under different management practices [44,45]. *R. irregularis* and *F. mosseae* are very productive members of the *Glomeraceae* family in terms of root colonization, nutrient foraging and association with other microbes. *R. irregularis* was dubbed an aggressive and quick colonizer of plants at low soil phosphorus (P) compared to many other species such as *Claroideoglomus claroideum* (*G. claroideum*) and *R. eutenicum* [46]. Since *R. irregularis* and *F. mosseae* are generalist mycobionts, they currently seem to be the best candidates to provide farmers and manufacturers maximum return on investment because they can be applied to varied hosts and survive long-term storage. Espinosa et al. (2005) [47] reported the effectiveness of seven species of AMF (*R. irregularis, R. fasciculatus, F. mosseae, R. clarus, Paraglomus occultum, Acaulospora scrobiculata and Diversispora spurcum*) evaluated using different tropical crops (potato, cassava, sweet potato, malanga, pepper, cucumber, tomato and banana), and found that *R. irregularis* performed consistently well for all crops evaluated. *F. mosseae and R. fasciculatus* also showed adequate performance in the same experiment. In another study, *R. irregularis* was found to be the most efficient in P foraging in patchy environments compared to *Gigaspora magarita and F. mosseae* [48]. Resistance of *F. mosseae* and *R. irregularis* to soil disturbance [46] makes them most suitable in both till and no-till fields.

Rouphael et al. (2015) [34] mentioned that for effective root colonization AMF inoculants should [49] (1) contain a mixture of AMF species, (2) a high number of propagules, (3) be free of pathogens, (4) include beneficial bacteria, and (5) must have a long shelf life. Although many of the examined products contain a mixture of AMF, not all of them contain a consortium of AMF, high propagule concentrations and beneficial bacteria. The commercial inoculants consist of two or more strains of AMF species, but a larger percentage is based on single strains only. Inoculants with a single species of AMF should not be considered inferior to those with multiple fungi species because some AMF single species (*F. mosseae*, *R. irregularis*) are generalists (capable of colonizing a large variety of host plants), have a longer shelf life, and are geographically distributed all over the world [43], thus these species may serve multiple functions and colonize multiple crops. Wagg et al. (2015) [50] also suggested that the composition of species within a consortium could be more important to improving productivity.

Apart from root colonization of host plants, AMF develop associations with vast communities of mycorrhyzospheric microbes that promote plant growth either by solubilizing P or secreting growth-promoting substances such as siderophore or indoleacetic acid (IAA) [51]. Our study shows that 19% of the inoculants comprise beneficial microbes while the rest do not. AMF inoculants consisting of consortium of P-solubilizing and N-fixing bacteria will be an added value for the industry. Moreover, long-term *in vitro* propagation of AMF has the potential to domesticate AMF species and alter their genetic functionalities, but co-culture with other microbes can help to mitigate putative genetic variation and function by activating AMF genes that would otherwise be silent or deleted [52]. Although inoculants containing beneficial microbes may confer many benefits to crops, more work needs to be done to characterize and isolate complementary growth-promoting bacteria for AMF inoculant production.

### 4.4. Propagule Concentration

The presence of viable propagules is important to the colonization of plant roots by AMF inoculants. After the AMF species composition, propagule concentration is the most interesting characteristic of commercial inoculants. Species composition may be more important to scientists but propagule number is a vital technical tool for the end users. Equivalent to nutrient composition of chemical fertilizers, the number of viable propagules may be used to determine the quantity of inoculants applied per hectare. After considering effectiveness of AMF species, it will be reasonable to expect inoculants with a higher propagule number to be more cost effective in terms of transportation costs and doses, leading to a better return on investment for manufacturers and growers. Our study showed that most products contain more than 100 propagules per gram, with 1 to 8 different AMF species. However, propagule concentration is independent of the number of AMF species.

### 4.5. Industrial Claims on the Effects of AMF-Based Inoculants

Nutrient uptake, stress alleviation, growth and crop quality enhancements are the dominant benefits attributed to most commercial inoculants (Figure 1). This is not surprising as many results from greenhouse and field trials have also proven the positive effects of AMF inoculation. AMF inoculation primarily boost nutrient mobilization and uptake [26,53,54,55,56,57]. AMF inoculation enhances uptake and transportation of nutrients such as P, N, Mg, K, Fe, Cu, Zn and Mo. [58,59,60]. Up to 75-90% P and 5-80% N uptake by mycorrhizal plants were attributed to AMF in the soil, but this amount may vary depending on crop species and field conditions [61,62]. Plant nutrition benefits are mainly attributed to the extraradical hyphae of AMF, which can spread to farther distances (where normal roots cannot reach), thereby increasing the surface area for nutrient uptake. For example, a study conducted by Karaca et al. (2013) [24] showed tremendous increase in soybean root and leaf growth when it was treated with a combination of AMF inoculant, phosphorus and sulfur as compared to the control. Uptake of macro- and micro-elements by AMF host plants is expected to result in improved growth, yield and crop quality. Stoffel et al. (2020) reported that inoculating maize crops (*Zea mays*) with AMF significantly increased P uptake, biomass and grain yield in low or medium soil P levels. Hijri (2016) [12] also demonstrated that inoculating potato crops with AMF inoculants increased average yield by 3.9 tons/ha, representing 9.5 % increment of total crop yield. 

Mycorrhiza inoculation especially in the field does not always produce positive effects on biomass and grain yield. For example, Faye et al. (2020) [22] showed that inoculating soybean (*Glycine max (L.) Merr*.) enhanced root nodulation but had no significant effect on grain yield. Similarly, Rosa et al. (2020) [21] showed that inoculation of grapevine rootstocks resulted in increased biomass production only in greenhouse but not in the field. N nutrition via AMF is still under debate. Bücking and Kafle (2015) [63] hypothesized that AMF are able to transfer N to their host plants through the mycorrhizal interface, but it has been suggested that higher N contents in inoculated plants is a consequence of the synergistic effect resulting from improved P uptake [64]. Additionally, Wang et al, (2018) [65] reported that AMF reduced the acquisition of N by plants in N limiting soils thereby affecting the plant health. AMF-mediated nitrogen acquisition was reported in some grass species but no effect was observed on total N uptake in the plant community [66]. Depression in growth of AMF inoculated crops in some cases may be linked to mutualism-parasitism-continuum whereby carbon cost of plant exceeds nutritional benefit obtained from AMF symbiosis, leading to increased cost to benefit ratio [45,62,64,67].

Climate stress alleviation in crops was also widely claimed by inoculant manufacturers. Smith et al. [68] mentioned that apart from the direct nutrition function, AMF also play a significant role in regulating the biochemical processes of a plant in the presence of abiotic and biotic stresses. Since abiotic stresses, such as drought, salinity, extreme temperature and trace elements, among others, have tremendous negative impacts on crop performance, AMF inoculants are becoming a very important tool to rely on considering the projected future consequences of climate change and environmental degradation. Liu et al. (2016) [25] reported that AMF inoculation of crops grown under lower temperature conditions had significant positive impact on the physiological features of the crops. Under drought stress, arbuscular mycorrhizal symbiosis promotes tolerance through stomatal control, direct water and nutrient uptake by the hyphae [69,70], adjustment of osmotic and antioxidant protection systems [71,72] and by increasing the regulatory function of the stress-responsive hormones [73]. In saline soils, mycorrhiza inoculation was reported to enhance plant survival through maintenance of cell homeostasis. Conversely, AMF effect under saline condition is not always positive; reduced viability and symbiotic function of AMF in extreme salt conditions have been reported. Romero-Munar et al. (2019) [74] reported reduced leaf growth and root colonization of *A. donax* (giant reed) inoculated with a commercial inoculant under moderate to high salt conditions.

Despite numerous claims by AMF inoculant dealers, economic gains from independent studies based on AMF inoculation remains unclear. Inoculation of crop with AMF often leads to increased root colonization but effect on yield is not always predictable despite improvement in plant nutrient composition and crop quality. For this reason, yield increase may not be the only criteria to justify the efficiency of bioinoculants. Moreover, AMF inoculant performance has not been consistent and depends on factors such as soil type, nutrient concentration, AMF species, crop genotype as well as biotic and abiotic stress conditions. As such, growers may need to identify crop type and operating environments including soil disturbance level and fertilization plan before selecting commercial inoculants. There may be need to choose single species inoculants which were suggested to be the best in controlled environment [75] or consortium of AMF species which is less host specific in the field. We present in Appendix A the results obtained in studies that evaluated some commercial AMF inoculants including some that were mentioned in our study.

### 4.6. Recommended Inoculant Application Methods and Doses

Soil application is a traditional method that has been practiced for decades and is more common than other bioinoculation pathways. According to Rocha et al. (2019) [76], direct soil application decreases physical damage to fragile seeds and cotyledons, minimizes the effects of pesticides and fungicides on the seeds, and gives smaller seeds the opportunity to be inoculated. Soil inoculation via liquid or powdered inoculants depends on the type of crops. It has been reported that powdered inoculants work best with grassy seeds such as wheat, barley, and oats, among others, because they are hairy textured, which means that the powder easily sticks to them [76,77]. On the other hand, liquid inoculants are often ideal for smooth-surfaced seeds such as corn, beans, and alfalfa because they form suitable adhesion. As per this survey, the application rates of inoculant products varied but application per hectare was commonly used. A range of 0.12 kg to 30 kg per hectare was considered as ideal depending on the strain composition and the crops they were applied to. This rate tallies with the quantity used in some studies conducted on coffee and horticultural crops, which showed an increase in dry matter yield at an application rate of 4–5 kg per hectare [2,78].

Seed treatment is a novel pathway for inoculant application and accounted for 26% of all the products surveyed. The main advantage of this technique is its ability to achieve immense precision in delivering the agents or active ingredients but the technique is faced with some challenges that may hinder its application and scaling up to commercial levels. Treated or coated seeds often lack uniformity in the quantity of inoculant received and may be contaminated by other microbes. Thick coatings may also hinder seed germination [79]. Selection and maintenance of viable inoculants in coated seeds are important areas that need to be addressed [80], especially for long-term storage. Moreover, due to lack of awareness in rural areas, the adoption rate among farmers remains low [17]. The doses for treating the seeds were a lot smaller, at maximum 2 g per 1 kg of seeds compared to soil treatment because only minimum inoculants are required for several seeds.

The least common application method reported in this study is the co-application of inoculants and chemical fertilizers, which only accounted for 11% of the total applications. Availability of P in the plant tissues resulting from application of chemical fertilizers may inhibit the colonization of roots by AMF [81,82]. In addition, excess P levels have been reported to cause toxicity to AMF. These reasons may limit large scale adoption of this method.

Cereal production clearly dominated the inoculation market space due to relatively greater quantities of cereal crops cultivated on large arable lands globally. Furthermore, during the early growth stages of production, P becomes a limiting nutrient for cereals, especially for root development, therefore, mycorrhizal-based inoculants would be vital in P acquisition [55]. Horticulture is the other major target market for several inoculant products. Inoculation of horticultural crops with AMF is increasing rapidly owing to improvement of the quality of the natural contents of inoculated crops. Rouphael et al. (2010) [83] reported that arbuscular mycorrhizal symbiosis can induce changes leading to the enhanced biosynthesis of phytochemicals, which are known to provide health benefits. Healthy dieting has led to increased demand for natural crops and this is likely to lead to a surge in inoculant production and usage.

## 5. Conclusions

This study is one of the foremost to examine commercial mycorrhiza inoculants extensively. Overall, the study shows that commercial AMF inoculants vary in terms of physical forms, species compositions, claims of functions, methods of application and recommendations. All the examined commercial products are based on Glomeraceae and are in three physical formulae, powder, granular, and liquid. Liquid inoculants are mainly based on single AMF species and have the least proportion among the inoculants while solid inoculants are more diverse in species composition and account for most of the AMF inoculants available. Many of the inoculants consist of other beneficial microbes, and this is believed to increase benefits and purposes served by these inoculants. AMF inoculant market is lopsided toward Asia, Europe, and the Americas. Production and application are still very low in developing countries especially Africa. Much still needs to be done to create awareness and investment to bring Africa among relevant players in the AMF inoculant industry.

In addition, the claims of AMF inoculant products provide at least three main benefits to plants; nutrient uptake, plant growth induction, and climate stress alleviation, which are extensively supported by scientific data. Most of these inoculants are applied to plants through the soil and very few coated on seeds. Most of these products are mainly applied to cereals; however, AMF inoculation faces the challenge of having short product shelf life, which often discourages long-term storage and transportation. Introduction of liquid inoculants is believed to address some drawbacks related to inoculant production and application, but the cost of liquid formulation is higher. Therefore, new technologies offering a longer shelf life such as seed coating, free drying and nano-encapsulation need to be examined. Seed coating is an emerging technology that has the potential for increased efficiency and is relatively cost effective due to minimal inoculum needed. Future biofertilizer research prospects should also focus on practical and cheap inoculum production based on a combination of *in vitro* AMF co-culture and co-inoculation with other plant beneficial microbes, such as plant growth-promoting bacteria (PGPB).

## Figures and Tables

**Figure 1 microorganisms-09-00081-f001:**
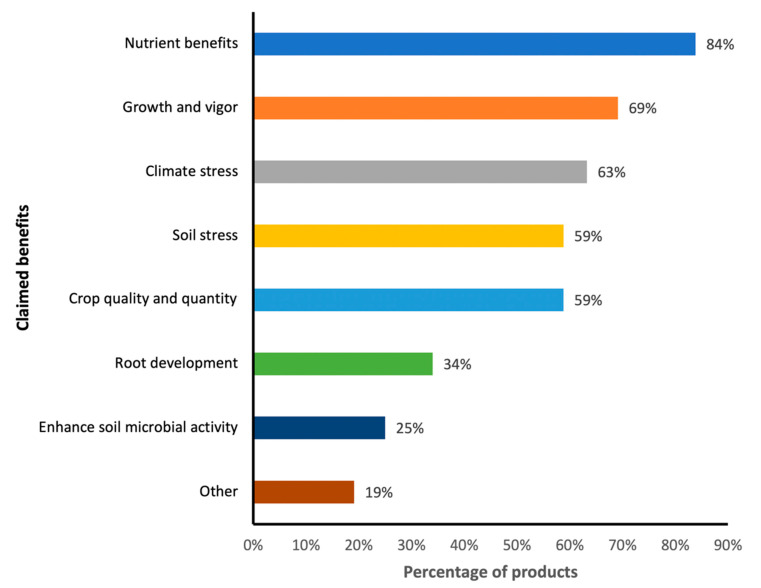
Percentage of the most frequent claims on inoculant effects on crop production.

**Figure 2 microorganisms-09-00081-f002:**
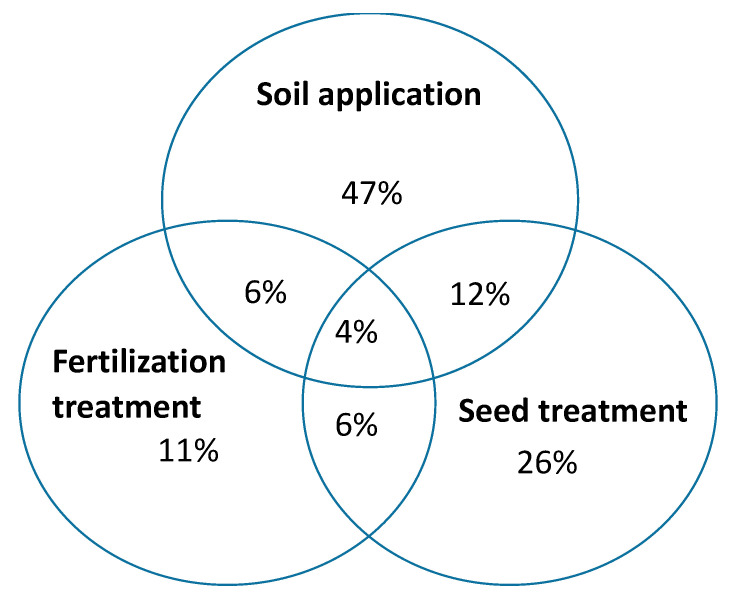
Application methods of the studied commercial mycorrhiza products.

**Figure 3 microorganisms-09-00081-f003:**
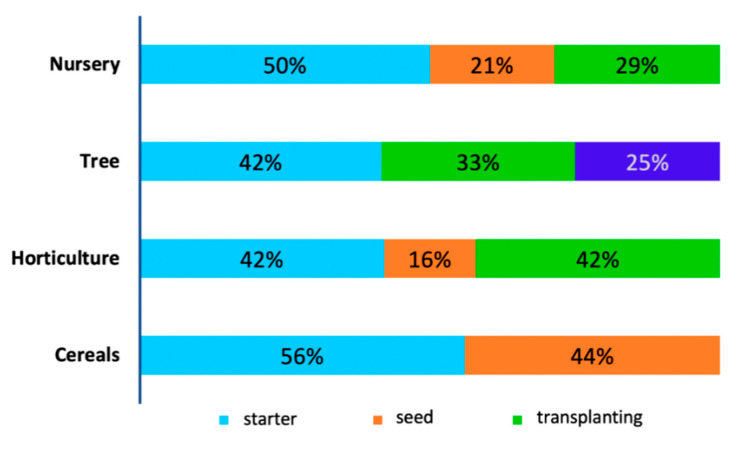
Percentages of application timing of the products for each field.

**Table 1 microorganisms-09-00081-t001:** Breakdown of products by company.

Company Name	Company’s Location	Product Name	Form	AMF Species
Ecological Resources, Inc./Oikos	USA/Chile	Oiko-Rhiza-E	powder	*R. irregularis, F. mosseae, Glomus Deserticola, R. clarus*
	USA/Chile	Oiko-Rhiza-Ectosol	powder	*Not specified*
Mycorrhizal Application, LLC/Sumitomo Chemical Chile S.A.	USA/Chile	MycoApply Endomaxx *	granular	*R. aggregatum, R. irregularis, F. mosseae, C. etunicatum*
Helena Agri-Enterprises, LLC	USA	Myco-Sol	powder	*R. aggregatum, R. irregularis, F. mosseae*
	USA	Myco-Vam Bare Root	powder	*R. aggregatum, R. irregularis, F. mosseae*
	USA	Myco-Vam Granular	granular	*R. aggregatum, R. irregularis, F. mosseae*
	USA	Myco-Vam Plus 6-3-3	powder	*R. irregularis, C. etunicatum, R. clarus*
Valent USA. LLC Agricultural Products	USA	MycoApply(R) Endo-Granular	granular	*R. aggregatum, R. irregularis, F. mosseae, C. etunicatum*
	USA	MycoApply(R) EndoFuse	Liquid	*R. aggregatum, R. irregularis, F. mosseae, C. etunicatum*
	USA	MycoApply(R) EndoMAXX	powder	*R. aggregatum, R. irregularis, F. mosseae, C. etunicatum*
	USA	MycoApply(R) EndoPrime SC(TM)	Liquid	*R. aggregatum, R. irregularis, F.s mosseae, C. etunicatum*
	USA	MycoApply(R) EndoPrime(TM)	powder	*R. aggregatum, R. irregularis, F. mosseae, C. etunicatum*
	USA	MycoApply(R) Ultrafine Endo	powder	*R. aggregatum, R. irregularis, F. mosseae, C. etunicatum*
Tainio Biologicals, Inc.	USA	MycoGenesis Seed Treatment	powder	*R. aggregatum, R. irregularis, F. mosseae, C. etunicatum*
	USA	MycoGenesis Soil Amendment	powder	*R. aggregatum, R. irregularis, F. mosseae, C. etunicatum*
JH Biotech, Inc.	USA	MYCORMAX Biological Inoculum	powder	*R. irregularis, F. mosseae*
	USA	MYCORMAX Biological Transplant Starter	powder	*R.s irregularis, F. mosseae*
AgroScience Solutions, LLC	USA	Organic Mycorrhizal Fungi	Liquid	*R. aggregatum, R. irregularis, F. mosseae*
Tainio Biologicals, Inc.	USA	Spectrum + Myco	powder	*R. aggregatum, R. irregularis, F. mosseae, C. etunicatum*
Sustane Natural Fertilizer, Inc.	USA	Sustane 3-7-2 with Mycorrhizae and Humates	granular	*R. aggregatum, C.s etunicatum, G. Deserticola, R. clarus*
Vegalab Inc.	USA	Vegalab MYCO BIOBOOST	powder	*R. irregularis*
Tainio Biologicals, Inc.	USA	Rhizogenesis	powder	*R. aggregatum, R. irregularis, F. mosseae, C. etunicatum*
Shemin Garden, LLC	USA	Ecofungi	powder	*R. aggregatum, R. irregularis, F. mosseae, C. etunicatum*
Pathway BioLogic, LLC	USA	Managefungi	powder	*R. aggregatum, R. irregularis, F. mosseae, C. etunicatum*
Symborg Inc.	Spain	MycoUp Activ *	powder	*R. iranicum*
	Spain	MycoUp Biological Inoculant *	powder	*R. iranicum*
Symborg	Spain	ResidHC *	powder	*R. iranicum*
	Spain	ResidMG *	granular	*R. iranicum*
Atens	Spain	Bio Asir Fruit	granular	*R. iranicum, F. mosseae*
	Spain	Aegis Sym irriga	powder	*R. iranicum, F. mosseae*
	Spain	Aegis Sym microgranulado	granular	*R. iranicum, F. mosseae*
Koppert	Netherlands	Panoramix *	Liquid	*Two AMF species*
INIFAP	México	Micorriza INIFAP	powder	*R. irregularis*
Plant Health Care	Mexico	PHC VAM.PWI	powder	*R. irregularis, C. etunicatum, C. etunicatum*
	Mexico	PHC ENDO-RHIZA	powder	*R. irregularis*
	Mexico	MycorTree-Injectable	powder	*R.s irregularis, C. etunicatum, R. clarus*
	Mexico	Turf Saber	powder	*R. irregularis, C. etunicatum, R. clarus*
	Mexico	Hotic Plus	powder	*R. irregularis, C. etunicatum, R. clarus*
OBA	Mexico	HIPER-GLOM	powder	*R. irregularis*
Vergel de Occidente	Mexico	Tec-Myc 60	powder	*AMF species not specified*
Biokrone	Mexico	Glumix	granular	*R. irregularis*
	Mexico	Glumix irigation	powder	*AMF species not specified*
Biofabrica Siglo XXI	Mexico	Micorrizafer plus	powder	*R. irregularis*
BIOMIC	Mexico	TM-73	powder, granular	*AMF species not specified*
Italpolina SpA/ATENS- Agrotecnologías Naturales S.L./Semillas Abe Ltd.a—In Pacta SpA	Italy/Spain/Chile	Aegis Gel *	powder	*R. irregularis, F. mosseae*
	Italy/Spain/Chile	Aegis Irriga *	powder	*R. irregularis, F. mosseae*
	Italy/Spain/Chile	Coveron	powder	*R. irregularis, F. mosseae*
GroundWork BioAG	Israel	Rootella G	granular	*R. irregularis*
	Israel	Rootella P	powder	*R. irregularis*
	Israel	Rootella F	powder	*R. irregularis*
	Israel	Rootella X	powder	*R. irregularis, F. mosseae*
	Israel	Rootella T	powder	*R. irregularis, F. mosseae, R. clarus*
	Israel	Rootella S	powder	*R. irregularis*
Agronutrition	France	CONNECTIS	Liquid	*R. irregularis*
MYCOSYM TRITION SL/Biosim	España/Chile	Mycosim Tri-Ton	granular	*R. irregularis*
PremierTech	Canada	Activ pulses granular	granular	*R. irregularis*
	Canada	Activ Soya granular	granular	*R. irregularis*
	Canada	Activ Soybean Powder	powder	*R. irregularis*
	Canada	Activ field crops granular	granular	*R. irregularis*
	Canada	Activ potato liquid	Liquid	*R. irregularis*
	Canada	Activ field crops liquid	Liquid	*R. irregularis*
	Canada	Activ field crops powder	powder	*R. irregularis*
	Canada	Activ specialty crops powder	powder	*R. irregularis*
	Canada	Activ specialty crops powder pea	powder	*R. irregularis*
Symborg Business Development S.L./Symborg Chile SpA	Chile and Spain	MycoUp *	granular	*R. iranicum*
	Chile and Spain	Resid HC *	granular	*R. iranicum*
Purely Organic Products LLC	USA	Pro-Yield Purely Pro N	__	*R. irregularis*
	USA	Pro- Yield Purely Pro P	__	*R. irregularis*

“*” indicates AMF inoculants that have independent testing results as shown in Appendix A.

## Data Availability

Not applicable.

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
