# Peer review of "Analysis of Arbuscular Mycorrhizal Fungal Inoculant Benchmarks"

_microorganisms, 2020, doi:10.3390/microorganisms9010081_

Round 1

Reviewer 1 Report

1. The Table and figures need considerable work. The data table could be expanded considerably to combine and include other information, for example country of origin, formulation, make up, etc. The figures could be move to the supplemental information as they currently stand.

2. The English and grammar could be improved throughout the manuscript.

3. The products used are not an exhaustive list and reference/mention of other reviews which list other products highlights this. This should be addressed.

4. The results and discussion could be combined.

5. Introduction could be tightened considerably.

Author Response

We are very grateful to the reviewer for taking the time to think about this work seriously and provide us with the opportunity to present a more concise paper. In view of the comments, we have modified the text particularly in results and discussion sections, added new Table S2 and references to address the reviewer's comments. We gave also added all information requested by the reviewer Table 1 This does not change the main conclusions of our study but soften some of them, which make the paper much stronger.

Response to Reviewer’s comments point-by-point are indicated bellow:

  1. The Table and figures need considerable work. The data table could be expanded considerably to combine and include other information, for example country of origin, formulation, make up, etc. The figures could be move to the supplemental information as they currently stand.

We have updated Table 1 buy adding  information on products, country of origin, product formulation and AMF species content.

  1. The English and grammar could be improved throughout the manuscript.

We have edited the English by a professional service.

  1. The products used are not an exhaustive list and reference/mention of other reviews which list other products highlights this. This should be addressed.

We agree with the reviewer's comment and added that information. We also added new Supporting table S2 which contains all references listed for AMF products.

  1. The results and discussion could be combined.

We thank the reviewer for this suggestion. We feel that it is better to leave the result and discussion separate.

  1. Introduction could be tightened considerably.

We have made changes to the introduction section and one paragraph was moved to discussion, see section 4.6.

Reviewer 2 Report

Dear Authors,

this work is focused on the evaluation and description of commercial products containing AMF. Biostimulants are certainly a topic of great interest in this period. At the moment, however, this work seems to me exclusively a baseline analysis of some commercial product on the market. I suggest some changes.
- Abstract: just as the actual role of these products does not emerge in the text. Have they been used in research activities? What action did they have? Please enter this information.
- Paragraphs 2.1 there are nine states and not ten.
- Table 1. Please add more details on the companies listed.
- The species all go in English.
- Information on the composition of all the components associated with commercial products is lacking. For example what other microbiies? Nutrients? etc ... What is meant by starter? For seed?
- Please extend this review activity with an evaluation of the end use of the product. What results have been achieved?

Author Response

We thank the reviewers for taking the time to read seriously our manuscript and provide us with the opportunity to present a more concise paper. We found comments of the reviewer very helpful to make this major revision of the manuscript that we feel answered all the criticisms and makes it a much more solid and rigorous manuscript.

Response to Reviewer’s comments point-by-point are indicated in Bold.

- Abstract: just as the actual role of these products does not emerge in the text. Have they been used in research activities? What action did they have? Please enter this information.

The term 'role' means claims or benefits, which was detailed in the text.

Most of commercial products mentioned in our study have been used in research activities in both greenhouse and filed trials. We have added the information in Table 1 as well as in a new supporting Tables S2 containing references of previous investigations and effects of AMF inoculants.

- Paragraphs 2.1 there are nine states and not ten.

We corrected the number to nine.

- Table 1. Please add more details on the companies listed.

We added the information of Country of origin, names of products, formulation and AMF species composition in the table 1.

- The species all go in English.

We corrected all names.

- Information on the composition of all the components associated with commercial products is lacking. For example what other microbiies? Nutrients? etc ... What is meant by starter? For seed?

We thank the reviewer for this comment. The list of other microbes in AMF inoculants can be found in the result section 3.2. We have added examples in that section: (for example: Bradyrhizobium japonicum, Bacillus sp., Azospirillum brasilense, etc.) and non-mycorrhizal fungal species (e.g. Trichoderma spp.)

- Please extend this review activity with an evaluation of the end use of the product. What results have been achieved?

We have extended our study by adding a new supporting table S2 and section in the Discussion to evaluate the performance of the mycorrhizal inoculants based on available information. References are also show in Supp. Table S2.

Round 2

Reviewer 1 Report

The manuscript could still be improved for language as some phrases used are not quite correct or common in their usage.

Figure 1 and 2 should still be removed. The supporting/additional information section would seem a good place for it. The text in the body of the manuscript can stay, but the figures 1 and 2 should go.

Author Response

We thank the reviewer for these suggestions which were taken into account in this revision.

Answers point by point

The manuscript could still be improved for language as some phrases used are not quite correct or common in their usage.

  • As you can see in the revised version, we have edited the English and rephrased many sentences for clarity.

Figure 1 and 2 should still be removed. The supporting/additional information section would seem a good place for it. The text in the body of the manuscript can stay, but the figures 1 and 2 should go.

  • We moved figure 1 and figure 2 to supporting documents. We also updated the order of figures and supporting figures.